# Estimation of Distances within Real and Virtual Dental Models as a Function of Task Complexity

**DOI:** 10.3390/diagnostics13071304

**Published:** 2023-03-30

**Authors:** Masrour Makaremi, Rafael Ristor, François de Brondeau, Agathe Choquart, Camille Mengelle, Bernard N’Kaoua

**Affiliations:** 1Equipe ACTIVE, BPH U1219, Inserm/University of Bordeaux, 146 rue Léo Saignat, CEDEX, 33076 Bordeaux, France; 2Department of Dentofacial Orthopedics, College of Health Sciences, University of Bordeaux, 146 rue Léo Saignat, CEDEX, 33076 Bordeaux, France

**Keywords:** distance perception, virtual systems, workload, orthodontics

## Abstract

Orthodontists have seen their practices evolve from estimating distances on plaster models to estimating distances on non-immersive virtual models. However, if the estimation of distance using real models can generate errors (compared to the real distance measured using tools), which remains acceptable from a clinical point of view, is this also the case for distance estimation performed on digital models? To answer this question, 50 orthodontists (31 women and 19 men) with an average age of 36 years (σ = 12.84; min = 23; max = 63) participated in an experiment consisting of estimating 3 types of distances (mandibular crowding, inter-canine distance, and inter-molar distance) on 6 dental models, including 3 real and 3 virtual models. Moreover, these models were of three different levels of complexity (easy, medium, and difficult). The results showed that, overall, the distances were overestimated (compared to the distance measured using an instrument) regardless of the situation (estimates on real or virtual models), but this overestimation was greater for the virtual models than for the real models. In addition, the mental load associated with the estimation tasks was considered by practitioners to be greater for the estimation tasks performed virtually compared to the same tasks performed on plaster models. Finally, when the estimation task was more complex, the number of estimation errors decreased in both the real and virtual situations, which could be related to the greater number of therapeutic issues associated with more complex models.

## 1. Introduction

Dental arch models are of great importance for orthodontists [1]. They are an essential part of diagnosis and are used to document the initial condition and to plan and measure the effects of therapeutic treatments. Models made using plaster casts obtained from impressions of the dental arch have made it possible to accurately represent dental arches and occlusions (how the mandibular teeth fit together with the maxillary teeth). Today, 3D digital models have become a dematerialized alternative [2]. These models are based on a set of anatomical data that are collected with diagnostic imaging equipment, that are processed by a computer, and that are displayed on a 2D monitor with depth perception methods to give a 3D image. The use of 3D models in orthodontics enables the fabrication of patient-specific 3D dental, facial, and/or skeletal arches for treatment planning, research, and forensic purposes [3]. The contributions of this technology to orthodontics are numerous and have been described many times [2,4,5]. They allow for reliable storage without the fear of loss or damage to the original castings [6]. They are also effective communication tools. Indeed, 3D digital models are a clear tool that allow practitioners to easily show their patients areas of deformation, levels of asymmetry, and the relationships between the different components of the face as well as the objectives and results of treatment, thus increasing patients’ understanding [3,6]. Compared to 2D recordings (photographs and X-rays), 3D digital models allow orthodontists to manipulate and extract relevant information directly from the 3D model without having to prolong the clinical examination or call the patient back [3]. With the ability to observe dental models with different occlusions and from different angles, virtual models allow for an increase in the accuracy of a practitioner’s examination of the intra- and inter-arch relationships and the transverse relationships between the upper and lower arches [6].

### 1.1. Objective Measurements of Distance Using Measurement Tools

Some studies have shown that when distances are measured using tools, the measurements made on plaster models or on digital models are all reliable and relatively similar [7,8,9]. These measurements are usually performed with a caliper for plaster models and with dedicated software (for calculating distances between two points) for digital models. A study by Radeke et al. [10] aimed to compare the traditional manual technique of using calipers to take orthodontic measurements on dental plaster casts versus a fully digital measurement technique based on the virtual 3D scans of casts. The results revealed no significant difference between the software-based and caliper-based measurements of mesiodistal tooth width. However, differences related to operator experience were observed. Inexperienced examiners took measurements faster using the software-based methods than when using a dental caliper.

Another study [7] showed similar results with the use of different measurement software and a scanner (cone beam computed tomography). Czarnota et al. [11], on the other hand, found differences between the two situations (the real model vs. the digital model) using a similar protocol (with the distances measured using tools), but they concluded that these differences were not clinically relevant. However, in the cited studies, distances were measured using measurement tools, and few studies have focused on the potential differences in distance estimation using virtual or physical models made by practitioners without measurement tools. This procedure (distance estimation without measurement tools) is nevertheless part of the clinical routine of professionals.

### 1.2. Subjective Measurements of Distance (Estimation) without Measurement Tools

If the distances actually measured do not differ significantly depending on whether the model is plaster or digital, it is important to ask about the subjective perception of distance (visual estimation without measurement tools). Indeed, the estimates of distance made in virtual environments and in their corresponding real environments do not seem identical. Most studies comparing these two situations focus on spatial navigation and the use of head-mounted displays (HMDs) to allow participants to physically move around in a virtual environment. They show that people underestimate the distances in virtual environments compared to those in the real world [12,13,14,15,16,17,18,19]. For example, Thompson et al. [18] experimentally manipulated the quality level of graphics in an immersive VE and found that all the levels were approximately 50% underestimated in a triangulated walking task (walking blindly towards a target previously seen along an initial oblique path and then turning towards the target) compared to the near 100% accuracy in a corresponding real-world environment.

Other studies have focused on estimating the distance between two points on the same object or estimating the distance between two objects. For example, Touranceau et al. [20] asked the participants in their study to estimate the distance between two points on the same object in an immersive 3D situation (stereoscopic display) and showed that the distance was globally overestimated by 10% compared to the same estimate made in a real environment.

Wartenberg and Wiborg [21] also asked the participants in their study to estimate the distance between two points on an object by comparing an immersive situation (a cube environment) to a non-immersive situation (a desktop environment). The results showed that, in the desktop environment, there was no significant prevalence of overestimation or underestimation. In the cube environment, however, overestimations were significantly more frequent than underestimations. Moreover, the desktop environment led to an increased magnitude of the estimation error, and this effect increased with the length of the distance to be estimated.

### 1.3. Aims

Distance estimation is central to the professional practice of orthodontists. However, in recent years, orthodontists have seen their practices evolve from estimating distances on plaster models to estimating distances on non-immersive virtual models. This development has led to new scientific questions within the orthodontic community [22,23], and the number of publications on this area is still limited due to the recent nature of this technological development. However, as we have seen, while the estimates made on plaster models can be close to the real distances (measured with tools), numerical models can generate larger biases in their distance estimations, which could have repercussions on the diagnoses and therapeutic decisions of practitioners.

In this context, the objective of our study is to compare the estimations of distance made for therapeutic purposes by practitioners on plaster models and on digital models. In both cases, the estimation biases are measured through comparisons with the real distances (measured using tools). The practitioner’s estimates of the mental loads associated with two example situations (a virtual estimate vs. a real estimate) are also compared. Finally, the models to be estimated are varied in complexity in order to be as close as possible to the clinical situations encountered by practitioners.

## 2. Materials and Methods

### 2.1. Subjects

A total of 50 orthodontists (31 women and 19 men) with a mean age of 36 years (σ = 12.84; min = 23; max = 63) participated in this experiment. Some of the participants were in the process of specializing in orthodontics at university (*n* = 23; first year = 7; second year = 7; third year = 9), and some were graduate dental surgeons and were specialized in orthodontics (*n* = 27). The training sites of the participants were universities in the cities of Bordeaux (*n* = 31), Rennes (*n* = 13), Toulouse (*n* = 3), Nantes (*n* = 1), Montpellier (*n* = 1), and Paris (*n* = 1). 

### 2.2. Material

To carry out this study, we used models of dental arches (in plaster and digital form) of three different levels of complexity and questionnaires which the participants filled in after completing the estimation tasks.

#### 2.2.1. Dental Arch Models

As a preamble to the research, we identified dental arch models of different levels of complexity (three levels of complexity were selected). For this, we used (and combined) three indices used in orthodontics to classify the difficulty of the cases [24]:The DHC (Dental Health Component): morphological component of the IOTN (Index of Orthodontic Treatment Need);The PAR Index (Peer Assessment Rating Index);The DAI (Dental Aesthetic Index).

These indices allowed us to classify the dental models into three levels of difficulty:Easy: anterior crowding only with contact point shift of 1 to 2 mm and without arch shape anomaly (normal inter-canine and inter-molar distances) or dental anomaly;Medium: anterior crowding resulting in a contact point shift of 2 to 4 mm with rotation or ectopia of one tooth per sector, asymmetry of less than 2 mm, and/or pathological arch shape;Difficult: anterior and medium crowding resulting in contact point shifts of more than 4 mm and/or ectopic positions and rotations of several teeth per sector, asymmetry greater than 2 mm, and/or pathological arch form.

The mandibular models selected were taken from the clinical database of an orthodontic practice in Bergerac (24100). After pre-selection, 64 files were selected, and each case was classified as easy, medium, or difficult. For each level of difficulty, two models were selected for their clinical interest (six models in total).

The virtual models were obtained by scanning the mandibular arch directly on the patients using a 3Shape Trios^®^ 3 intra-oral camera. The plaster models were obtained by printing virtual models with a NextDent 5100^®^ DLP (Digital Light Processing) printer using Sprint Basic^®^ (2020.12) 3D software and then making alginate impressions of these printed plastic models. The Figure 1 shows examples of a real mandibular model (Figure 1a) and a virtual model (Figure 1b).

#### 2.2.2. Questionnaires

Various questionnaires were submitted to each participant.

##### Demographic Data

Prior to the completion of the tasks, a demographic data collection questionnaire was completed by the participants (including gender, age, year of starting orthodontic practice, year of residency if they were interns, place of training/specialization, and use of virtual and/or real models in their training/profession).

A questionnaire was also developed to collect the participants’ experience and computer skills (Appendix A).

##### Mental Load

After each model was presented, the participants had to evaluate the mental load associated with the estimation task they had to perform. This allowed us to obtain an estimate from the participants of the mental load associated with each of the estimates made (depending on the type of model—plaster vs. virtual—and the level of complexity).

The NASA-TLX (Appendix B), which we used in this study, is a questionnaire that was developed in the 1980s by the Human Performance Group at NASA-Ames Research Center, CA (Sandra Hart). It assesses six dimensions of a task: mental demand, physical demand, time demand, performance, effort, and frustration. These dimensions are each presented in the form of a Likert scale (ranging from 0 to 20), which provides a score for each dimension and an overall score per participant. The questionnaire was presented in its simplified version [25]. It has been validated for mental workload assessments in the healthcare sector [26].

### 2.3. Procedure

The first phase included the participants signing a consent form and the researchers providing a description and an explanation of the task to them. The participants were also invited to complete a questionnaire regarding their experience on a tablet computer. This questionnaire was presented on the tablet in the form of a Google Forms questionnaire.

During the second step, the subjects had to estimate the distances on the dental models presented. The order of presentation of each model, the order of presentation of the levels of complexity, and the choice of the models presented in the real and virtual environments were previously established randomly. The digital models were presented using a computer with a traditional 17-inch 2D monitor (DELL-TRPO030351 Intel(R) Core (TM) i7-7700HQ CPU 2.80 GHz) without a 3D stereo image. The participants were able to manipulate them using a 2D mouse on OrthoViewer^®^ (2021.1) software. The plaster models were presented on a white desk and could be manipulated by hand by the participants. After each model was presented, the participants were asked to complete the NASA-TLX questionnaire. They completed the questionnaire six times.

For each model, the three distances to be estimated were mandibular crowding, inter-canine distance, and inter-molar distance.

Crowding is an intra-arch parameter which can be assessed on the maxillary and mandibular arches. The estimation of this distance is important in the evaluation of dento-maxillary disharmonies. In our study, we asked the participants to perform a crowding estimate only on the mandibular arch and on the 10 anterior teeth (36 mesial to 46 mesial);The inter-canine distance is the right transverse distance measured between the top of the right and left canine tips;The inter-molar distance is the straight transverse distance joining the apexes of the mesio-vestibular cusps of the first permanent molars.

The Figure 2 shows the three distances (red lines) studied.

The first two estimates (mandibular crowding and inter-canine distance) are part of the diagnoses classically made by professionals. The inter-molar distance, on the other hand, is a control condition that is never estimated in practice by professionals.

It is important to note that the objective measurements (using the measuring instruments) of these three distances (mandibular crowding, inter-canine distance, and inter-molar distance) were made with Ortho Analyzer^®^ software for the digital models and using a foot with a digital slide and a brass wire (traditional measurement) for the plaster models. This operation was performed by two different operators.

### 2.4. Statistical Analysis

The results were analyzed by using two-factor ANOVA with repeated measures. For each ANOVA, the two within-subject factors were the “presentation” factor (real vs. virtual) and the “complexity” factor (three levels). The three dependent variables (quantitative) corresponded to the three distance estimates made on each of the models (mandibular crowding, inter-canine distance, and inter-molar distance). More precisely, for each situation, the dependent variable corresponded to an indicator which was the error score = distance estimated by the practitioner—real value (the distance measured using an instrument). The distance was overestimated when the indicator was greater than 0 and was underestimated when the indicator was less than 0. Each participant passed the six conditions (three levels of complexity x two presentation conditions), and the order of presentation of the conditions was counterbalanced.

Mental workload was also analyzed using two-factor ANOVA with repeated measures. As for the previous analyses, the within-subject factors were presentation (real vs. virtual) and complexity (three levels). The dependent variable was the score of the NASA-TLX questionnaire.

Before each ANOVA, the samples were subjected to a test of normality (Shapiro’s test) as well as to a test of the homogeneity of variances (Levene’s test). These analyses indicated that all the data followed normal distributions and that the variances were homogeneous. In accordance with past studies in the field, this research adopted a significance level (α) of 0.05 and, hence, a confidence level of 95% (1 − α).

## 3. Results

### 3.1. Mental Load

The results of the two-factor ANOVA showed the main effect of the following factors:The type of presentation (real vs. virtual) (F (1, 49) = 16.778, *p* < 0.05): the descriptive results (Table 1) show that, for the participants, the mental load associated with estimating distances was higher for the virtual models than for the real models (all levels of complexity combined);The level of complexity (F (2, 98) = 5.331, *p* < 0.05): the descriptive results (Table 1) show that, for the participants, the mental load associated with estimating distances was the lowest in the easy condition, then increased in the medium condition, and was highest in the difficult condition (for the real and virtual conditions).

Finally, the analysis of variance did not show any significant complexity x presentation interaction (F (2, 98) = 1.342, *p* < 0.05).

The analysis therefore shows that, for practitioners, the mental load associated with the estimation task was greater in the real situation than in the virtual situation and that, for these two situations, the mental load increased with the complexity of the models being evaluated.

### 3.2. Distance Estimates

For each model presented, the participants had to estimate two distances (mandibular crowding and the inter-canine distance) which correspond to the situations encountered in the professional practice and a distance (the inter-molar distance) which is never evaluated in the professional setting. The results obtained on the distance estimates, according to the presentation (virtual vs. real) and the complexity, were identical for the first two distances and differed for the third distance. The mean and standard deviation of the estimation error of each distance are presented in Table 2.

#### 3.2.1. Mandibular Crowding and Inter-Canine Distance

The analysis of variance (Table 3) showed the main effects of the following factors:The type of presentation (real vs. virtual) (F (1, 49) = 7.662 and *p* < 0.05 for mandibular crowding and F (1, 49) = 6.053 and *p* < 0.05 for the inter-canine distance): these two distances (Table 2) were overestimated both in reality and in the virtual environment, and this overestimation was greater in the virtual conditions than in the real conditions;The level of complexity (F (2, 98) = 5.139 and *p* < 0.05 for mandibular crowding and F (2, 98) = 4.078 and *p* < 0.05 for the inter-canine distance): for these two estimations (Table 2), the overestimation was smaller in the difficult condition than in the easy and medium conditions.

**Table 3 diagnostics-13-01304-t003:** Results of the two-factor ANOVA (type of presentation and level of complexity) of the three estimated distances (mandibular crowding, inter-canine distance, and inter-molar distance).

	Modality	F	*p*
Mandibular Crowding	Presentation	7.662	0.008 *
	Complexity	5.139	0.008 *
	Interaction	0.951	0.390
Inter-Canine Distance	Presentation	6.053	0.017 *
	Complexity	4.078	0.020 *
	Interaction	3.154	0.047 *
Inter-Molar Distance	Presentation	0.441	0.510
	Complexity	1.740	0.181
	Interaction	1.660	0.195

* *p* < 0.05.

Finally, we did not observe an interaction between the presentation type and complexity level (F (2, 98) = 0.951, *p* > 0.05).

The results therefore show an overestimation of these two distances in both the real situation and the virtual situation (compared to the distances actually measured), but this overestimation was significantly greater in the virtual situation compared to the real situation.

#### 3.2.2. Inter-Molar Distance

The analysis of variance (Table 3) showed no main effect of the type of presentation (real vs. virtual) (F (1, 49) = 0.441, *p* > 0.05) or the level of complexity (F (2, 98) = 1.740, *p* > 0.05) and no interaction between presentation and complexity (F (2, 98) = 1.660, *p* > 0.05). In this situation and unlike the two other situations, the estimates did not vary according to the type of presentation (real vs. virtual) nor according to the level of the complexity of the estimate to be produced.

#### 3.2.3. Effect of Experience

We also tested the effect of the practitioners’ orthodontic experience on their distance estimation using a linear regression model. This analysis did not show any significant correlation between the estimation errors and the orthodontal experience of the practitioners (r = −0.14; *p* = 0.31). Estimation errors therefore do not vary according to the professional experience of the practitioner.

## 4. Discussion

Orthodontal practitioners rely on distance estimates to carry out diagnostics and therapeutic follow-ups. Indeed, previous research conducted in the UK [27] showed that, while most orthodontists learn formal analysis (using measurement tools) during their training, relatively few continue to use it once they start to work as a specialist orthodontist. The possible reasons for this could be the time and complexity required to perform such a formal analysis and the idea that, with professional experience, diagnostic decisions through direct visualization (estimates) are equally as effective.

However, when these estimations were carried out on plaster models, it was shown that the estimated distances were slightly different from the real distances (measured using a measurement tool) [27,28,29].

Indeed, in their study, Wallis et al. [27] showed that there could be not only large disparities in the estimation of crowding between practitioners but also over time for the same practitioner. In the study, the crowding estimates varied by up to 15 mm for the same model between orthodontists and by up to 3 mm by the same orthodontist over time. In contrast, despite these disparities in their estimates, there was less variation in the treatment decisions regarding whether or not to treat patients with extractions. Another study [29] showed that direct visualization (distance estimation) can lead to an overestimation of crowding. When the true amount of crowding is known, it can lead to more consistent treatment planning, with the decision being made to extract fewer teeth in borderline cases. These previous studies show that distance estimations carried out for therapeutic purposes can be more or less precise when they are carried out on plaster models, but such disparities only marginally affect treatment decisions.

However, today, digital models have replaced plaster models in the orthodontic practice. Only one study has focused on the use of numerical models [30], and it showed an overestimation of the mandibular crowding distances estimated by 31 orthodontists when the estimations were made on virtual models.

However, no study has directly compared the distance estimates made by practitioners on plaster models and on virtual models in order to measure the impact of the change in practice on estimation biases and therapeutic decisions. This comparison is the main objective of our study.

One of the main results of our study is not only that distances are overestimated when estimations are performed both on plaster and digital models but also that overestimation is significantly greater on digital models. Very few studies in the literature have compared the distance estimations (between two points or two objects) carried out in virtual environments and real environments. However, these studies showed that distances were globally overestimated when they were estimated in virtual environments [20]. Moreover, a non-immersive environment (a desktop environment) led to an increased magnitude of the estimation error compared to an immersive environment, and this effect increased with the length of the distance to be estimated [21]. In the orthodontic practice, overestimation is therefore accentuated by the fact that digital models are presented in non-immersive (desktop) environments. Another result of our study is that, when practitioners were asked to evaluate the load related to the task to be performed, they indicated a higher mental load to perform estimates on virtual models than on real models. As a result, the estimation error was greater in the virtual situation, and the participants considered (with regard to the results of the NASA-TLX questionnaire) that the mental load associated with estimating distances in the virtual situation to be greater than that associated with the same task in the real situation. These results are consistent with the literature, showing that a higher mental workload is associated with greater task complexity [31,32,33,34] and that a high mental workload is generally associated with a decrease in performance [35]. In our study, the decrease in performance corresponded to an increase in the distance estimation error in a virtual situation.

Another interesting result of our study is that the difference between real and virtual situations in the estimation of distance only appeared for the two distances that had therapeutic significance for the practitioner (mandibular crowding and the inter-canine distance). This difference no longer appeared for the estimate of the distance which did not appear in usual practice (the inter-molar distance).

Moreover, our results (Table 3) indicate that the estimation errors (real and virtual situations combined) were at their maximum value for the estimation of the inter-molar distance (3.52) compared to the estimation of the other two distances (mandibular crowding: 2.11) (inter-canine distance: 3.05). This result is important because it allows us to conclude that, for practitioners, the distances that they are used to estimating in the clinical practice (in reality or a virtual environment) are associated with fewer significant estimation biases than the distances that they are not used to estimating.

Our results also show that the complexity of the model has an impact on distance estimation. Indeed, an increase in the complexity of the models (both in the virtual environment and in reality) was associated with better distance estimates for mandibular crowding and the inter-canine distance, whereas no difference in the estimates according to task complexity was only found for the inter-molar distance. Such a result could then be explained by the therapeutic issues related to the distances to be estimated. Indeed, mandibular crowding and the inter-canine distance have an important role in patient diagnosis and, therefore, the treatment that will be put in place, whereas the inter-molar distance is a condition that is not encountered in the professional practice. Studies have shown that task goals mediate the relationship between performance and complexity. Important goals associated with complex tasks can improve performance through motivational and cognitive processes which lead to the development of effective strategies [36,37]. Thus, in our study, estimating distance in complex cases (for estimates that correspond to real-life therapeutic situations) may be associated with more severe surgery problems or potentially more impactful subsequent management. These higher stakes when estimating complex models could be the reason for the increased attention in estimating distance, thus improving performance. Such a hypothesis obviously remains to be tested in further investigations, particularly in terms of the impact of the therapeutic challenge on the accuracy of the estimation responses in real and virtual environments.

Finally, in our study, the importance of the estimation errors was not correlated with the professional experience of the practitioners. The same result was shown by Wallis et al. [27] on plaster models. The authors observed great variability in the practitioners’ estimations, but the level of the experience of the orthodontists did not affect the precision of their estimation of the severity of crowding.

## 5. Conclusions

Our results show that the errors in the distance estimations (compared to the real distances measured using a measuring instrument) made by practitioners are greater when the estimations are carried out on virtual models as opposed to real models. In addition, the mental load associated with estimation tasks is considered by practitioners to be greater for estimation tasks performed virtually compared to the same tasks performed on plaster models. Finally, when the estimation task is more complex, the estimation errors decrease in both real and virtual situations, which could be related to the greater therapeutic issues associated with more complex models.

Such a study comparing virtual situations (digital models) and real situations (plaster models) has not been conducted thus far and could have implications that are both theoretical (a better understanding of the distance estimation biases made in non-immersive virtual environments compared to real situations) and practical (taking into account these estimation biases in the diagnoses made by professionals). On a clinical level, the results of this study could allow for better training of practitioners, particularly in new practices using digital tools (by providing more systematic information on the possible biases of estimates linked to digital tools), or could even allow for devices to be designed that are capable of correcting these possible estimation biases.

## Figures and Tables

**Figure 1 diagnostics-13-01304-f001:**
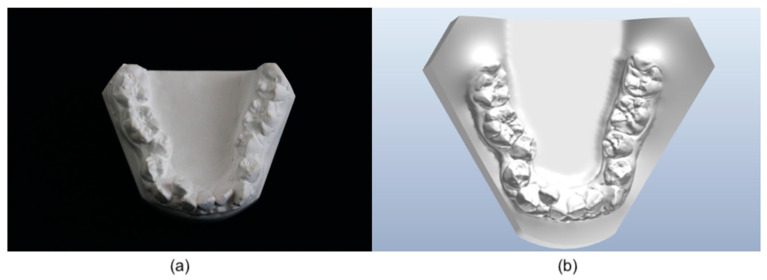
Example of (**a**) a real mandibular model and (**b**) a virtual mandibular model rendered by Ortho Analyzer^®^ (2021.1) software.

**Figure 2 diagnostics-13-01304-f002:**
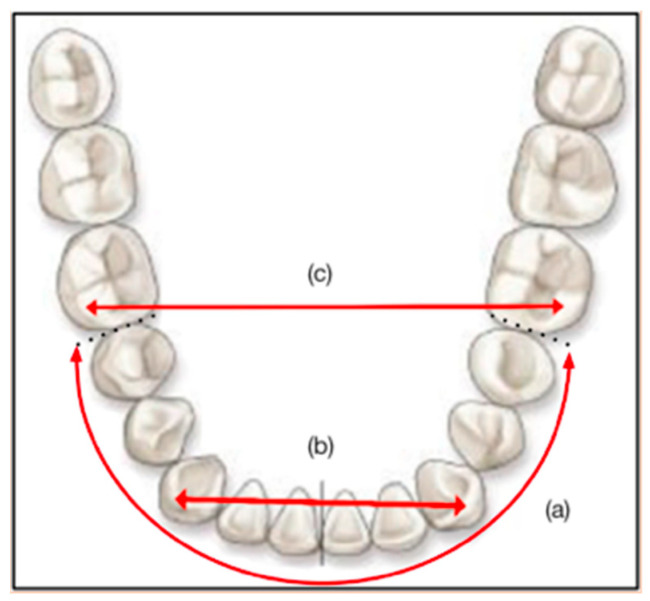
Indicative diagram of (**a**) mandibular crowding, (**b**) inter-canine distance, and (**c**) inter-molar distance (red lines).

**Table 1 diagnostics-13-01304-t001:** Descriptive statistics (M = mean; σ = standard deviation) of mental load by level of complexity and type of presentation (real or virtual).

Mental Load						
Level of Complexity	Easy		Medium		Difficult	
	M	σ	M	σ	M	σ
Virtual	44.9	17.68	47.4	19.19	51.42	20.06
Real	51.8	18.73	53.24	17.67	54.74	18.86

**Table 2 diagnostics-13-01304-t002:** Descriptive statistics (M = mean of the estimation errors; σ = standard deviation) of crowding, inter-canine distance, and inter-molar distance estimation in virtual and real conditions.

	Mandibular Crowding						Inter-Canine Distance						Inter-Molar Distance					
Level of Complexity	Easy		Medium		Difficult		Easy		Medium		Difficult		Easy		Medium		Difficult	
	M	σ	M	σ	M	Σ	M	σ	M	Σ	M	σ	M	σ	M	σ	M	σ
Virtual	2.61	2.18	2.34	2.76	1.62	4.16	3.35	8.75	1.83	7.93	2.48	8.76	3.13	9.79	3.82	9.03	2.83	11.12
Real	2.84	2.54	3.29	2.95	2.17	4.21	4.71	9.08	3.69	8.88	2.25	8.27	4.84	10.67	4.39	8.10	2.13	9.32

## Data Availability

The data is unavailable due to privacy.

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
