# Peer review of "Estimation of Distances within Real and Virtual Dental Models as a Function of Task Complexity"

_diagnostics, 2023, doi:10.3390/diagnostics13071304_

Round 1

Reviewer 1 Report (Previous Reviewer 1)

The manuscript "Estimation of distances within real and virtual dental models as a function of task complexity" deals with a very important topic. 

However, there are some issues that need to be further analyzed: 

The abstract is too long and confusing. 

The introduction is too long and should be better structured. References are not in journal style. 

The aim part of the study is overwritten (1 page!) and confusing. 

The material and method section is poorly described, although the authors used long and complicated sentences. 

The results are not presented clearly, it is very difficult to follow the aims and the resuts of the study. 

The discussion should focus on comparing the results to similar ones from the literature. 

The conclusion is too vague and not supported by the results. 

References are not in journal style

The English language must be mproved

Author Response

Reviewer 1

[General comment] The manuscript "Estimation of distances within real and virtual dental models as a function of task complexity" deals with a very important topic. However, there are some issues that need to be further analyzed: 

Thank you very much for your comments. The article has been rewritten considering the suggestions of the reviewers.

[Comment 1] The abstract is too long and confusing. 

The summary has been reduced and largely clarified.

[Comment 2] The introduction is too long and should be better structured. References are not in journal style. 

The introduction has been shortened and better structured around three major points: the use of digital models in orthodontics, the objective measurements of distances using measurement tools, and the subjective measurements of distance (estimation) without measurement tools. The references are now in MDPI journal style.

[Comment 3] The aim part of the study is overwritten (1 page!) and confusing. 

The aim part has been shortened and clarified.

[Comment 4] The material and method section is poorly described, although the authors used long and complicated sentences. 

In order to make this section more understandable, the material and method part has been rewritten.

[Comment 5] The results are not presented clearly, it is very difficult to follow the aims and the results of the study. 

The results have been rewritten, simplified and more clearly linked to the aims.

[Comment 6] The discussion should focus on comparing the results to similar ones from the literature. 

The discussion has been completely rewritten and the results are now compared to those in the literature.

[Comment 7] The conclusion is too vague and not supported by the results. 

The conclusion has been rewritten in line with the results.

[Comment 8] References are not in journal style

References are now in MDPI journal style.

[Comment 9] The English language must be improved

The English language has been proofread and corrected by professional translators.

Reviewer 2 Report (Previous Reviewer 2)

I think the authors sent us a wrong version, because: 1. The version is not following MDPI style 2. There are lots of corrections and suggestions 3. the REferences are not following MDPI style 4. Figure 2 is an original figure? 5. Other language is used in the manuscript (french), the manuscript should be in english 6. The organization of the paper seems like a scholarship report and not an article 7. The idea of the article is good but please put the paper in MDPI format and correct all the references in the text and reference liste following MDPI style The authors should revise MDPI recommendations I will give the other a second chance, i will ask for major revision

Author Response

Indeed, it was a bad version. The article has been completely rewritten following the MDPI recommendations. The English language has been proofread and corrected by professional translators. Many thanks to the reviewer for giving us a second chance.

Reviewer 3 Report (Previous Reviewer 3)

The authors have managed to address most of the raised issues in my previous review of the paper. I think the current submission is better than the previous one. I believe that this work would add valuable information to the existing literature.

Author Response

Thank you very much.

Round 2

Reviewer 1 Report (Previous Reviewer 1)

The manuscript has been improved. 

Reviewer 2 Report (Previous Reviewer 2)

Good answers

This manuscript is a resubmission of an earlier submission. The following is a list of the peer review reports and author responses from that submission.

Round 1

Reviewer 1 Report

Estimation of Distances within Real and Virtual Dental Models as a Function of Task Complexity

The title is misleading and should be changed – to better reflect the study’s aim

The abstract is too long – 200 words should be pursued

Keywords – Mesh terms should be used

The introduction is too long and describes many other uses than for orthodontics itself, it should be better focused on the topic. Physical models and Virtual models should describe only for orthodontic purposes

Lines 55-67 should be rethought also, and 69-83

Distance estimation in non-immersive environments should be explained as why authors decided to discuss this topic in the introduction

The aims, 1.6 should be better explained in a shorter manner

Methods: ethical approval number is required.

The methodological reasons should have a reference.

Figure 1 should describe also the software used

The Questionnaires should be presented as a separate annex

Was it a validated Questionnaire? Specificity should be provided

The Mental load should be defined, and a figure provided

The 2.3 Procedure should be described in a concise manner

For figure 2 a reference should be provided, unless it is an original image

Figure 3 is not necessary

The statistical analysis and tests should be provided in the tables, not in the text

Figure 4 is unclear – it should be better described in the text and removed

3.2.1-3.2.4 should be better presented as a table – it is difficult to follow

The real clinical relevance should be provided

What are the limitations?

AOC should be provided

NNT as well

The conclusion should be focused on the research, not on future directions. It is vague

Overall: what is the originality of this study???

References are not in journal style

Reviewer 2 Report

Abstract: good   Introduction: - Please correct all the reference numbers in the text
- the originality is clear but please clarify the aim of the present study   Methods: - What about the experience level of the participants? 7 from 1st year, 7 from the second year... could this difference influence the results? - Please add an example of the questionnaires format - Please define the statistical significance level    Results: - Please mention table 1 into the text and add the statistical significance - Line 346: Figure 5? Where is figure 5? - Please mention the figure 4 in the text - Please re-organize the appearance of all figures and table in the manuscript - Please modify the references following MDPI style   The article should be strongly re-organized, the appearance of the references should be revised The article could not be accepted in the current form but the idea is good I have to reject the article and encourage the authors to make a re-submission once the article has a good organization

Reviewer 3 Report

Thanks for allowing me to review this manuscript. I think that the paper is well written. 

I have the following points that need to be addressed before this paper can be considered for publication.

Introduction

1- I have some concerns regarding the length of the Introduction section. Some parts of this section can be reduced to a certain extent. I believe the authors should go directly to the core of their topic. In other words, they should focus on what is related to orthodontics and maxillofacial surgery.

1- In line 57, please consider adding this citation to this given statement since this paper discusses the advantages of 3D models in surgical planning: Hajeer MY, Millett DT, Ayoub AF, Siebert JP. Applications of 3D imaging in orthodontics: part I. J Orthod. 2004 Mar;31(1):62-70. doi: 10.1179/146531204225011346. PMID: 15071154.

2- In line 89, please consider adding this citation to the given statement since this paper deals exactly with the main topic of this part of the introduction: Hajeer MY, Millett DT, Ayoub AF, Siebert JP. Applications of 3D imaging in orthodontics: part II. J Orthod. 2004 Jun;31(2):154-62. doi: 10.1179/146531204225020472. PMID: 15210932.

Materials and Methods

3- In line 175, how did you choose your assessors? Why did you include dental surgeons? This mixture is not helpful.

4- In line 226, why did you not allow the participants to set the dental arch image in 3D?

Results

They are presented in a very good manner.

Discussion

5- In the section, please compare and discuss your results in light of the following studies relevant to two points (1: the use of 3D printing in orthodontics; 2: the accuracy of orthodontic measurements on 3D models):

1: Jaber ST, Hajeer MY, Khattab TZ, Mahaini L. Evaluation of the fused deposition modeling and the digital light processing techniques in terms of dimensional accuracy of printing dental models used for the fabrication of clear aligners. Clin Exp Dent Res. 2021 Aug;7(4):591-600. doi: 10.1002/cre2.366. Epub 2020 Nov 30. PMID: 33258297; PMCID: PMC8404487.

2: Hajeer MY. Assessment of dental arches in patients with Class II division 1 and division 2 malocclusions using 3D digital models in a Syrian sample. Eur J Paediatr Dent. 2014 Jun;15(2):151-7. PMID: 25102466.

Conclusion

They are fine.